# Treatment Costs of Colorectal Cancer by Sex and Age: Population-Based Study on Health Insurance Data from Germany

**DOI:** 10.3390/cancers14153836

**Published:** 2022-08-08

**Authors:** Thomas Heisser, Andreas Simon, Jana Hapfelmeier, Michael Hoffmeister, Hermann Brenner

**Affiliations:** 1Division of Clinical Epidemiology and Aging Research, German Cancer Research Center (DKFZ), 81673 Heidelberg, Germany; 2Medical Faculty Heidelberg, University of Heidelberg, 69120 Heidelberg, Germany; 3Vilua Healthcare GmbH, 81673 Munich, Germany; 4Division of Preventive Oncology, German Cancer Research Center (DKFZ) and National Center for Tumor Diseases (NCT), 69120 Heidelberg, Germany; 5German Cancer Consortium (DKTK), German Cancer Research Center (DKFZ), 69120 Heidelberg, Germany

**Keywords:** colorectal cancer, screening, healthcare costs, economic burden, cost-effectiveness

## Abstract

**Simple Summary:**

Screening for colorectal cancer (CRC) has considerably contributed to declining CRC incidence and mortality rates in Germany over the last two decades. However, evidence on the cost-effectiveness of this public health remains scarce, mostly due to the lack of detailed sex- and age-specific estimates on CRC treatment costs over time. Using a large research database on insurance claims data, we analyzed CRC-related inpatient, outpatient as well as medication costs up to five years after diagnosis and prior to death. Our findings show that costs in the terminal phase of care were consistently on a high level even several years preceding death, mostly driven by high inpatient and medication costs, and substantially higher as compared to the initial phase of care. As well, we observed a consistent pattern of higher costs in men versus women, most markedly in the first year of diagnosis and the final year of life, and strongly varying costs by age. Our findings could be highly useful in informing cost-effectiveness analyses e.g., to further optimize strategies to screen for CRC.

**Abstract:**

**Objective:** Evidence on the cost-effectiveness of screening for colorectal cancer (CRC) in the German general population remains scarce as key input parameters, the costs to treat CRC, are largely unknown. Here, we provide detailed estimates on CRC treatment costs over time. **Methods:** Using insurance claims data from the Vilua healthcare research database, we included subjects with newly diagnosed CRC and subjects who died of CRC between 2012 and 2016. We assessed annualized CRC-related inpatient, outpatient and medication costs for up to five years after first diagnosis and prior to death, stratified by sex and age. **Findings:** We identified 1748 and 1117 subjects with follow-up data for at least 1 year after diagnosis and prior to death, respectively. In those newly diagnosed, average costs were highest in the first year after diagnosis (men, EUR 16,375–16,450; women, EUR 10,071–13,250) and dropped steeply in the following years, with no consistent pattern of differences with respect to age. Costs prior to death were substantially higher as compared to the initial phase of care and consistently on a high level even several years before death, peaking in the final year of life, with strong differences by sex and age (men vs. women, <70 years, EUR 34,351 vs. EUR 31,417; ≥70 years, EUR 14,463 vs. EUR 9930). **Conclusion:** Once clinically manifest, CRC causes substantial treatment costs over time, particularly in the palliative care setting. Strong differences in treatment costs by sex and age warrant further investigation.

## 1. Introduction

Since the introduction of screening colonoscopy in Germany in the year 2002, the incidence and mortality of colorectal cancer (CRC) have declined by more than 20% and more than 35%, respectively [1], and now approximately 50% of incident cancers are detected at stages I or II, amendable to curative intent treatment [2]. Reflecting on these better prospects for the cure of earlier tumor stages, it has previously been shown that patients with screen-detected CRC have much better prognosis than patients with symptom-detected cancer [3].

However, even though screening has been remarkably successful in reducing the CRC burden, evidence on the cost-effectiveness of this public health measure in the German population remains scarce [4,5,6]. This may largely be attributed to the lack of key determinants for such an analysis, most importantly detailed estimates on CRC treatment costs. The few data available in this respect [6,7] do not allow to group patients by essential parameters such as sex and age at diagnosis, which are likely to impact CRC treatment costs. Previous evidence from other countries suggests that males may have higher treatment costs than females for the majority of cancer types [8,9,10], and within cancer site and stage at diagnosis, younger cancer patients tend to receive more (cost-) intensive therapy than older patients [11,12,13]. In addition, little is known about the costs of CRC over time, in particular those arising beyond one year of diagnosis and preceding the final year prior to death. The availability of such stratified and more detailed cost metrics would, firstly, allow to more exactly quantify the burden of disease from a health-economic point of view [14]. Secondly, sex- and age-specific CRC treatment costs could inform and improve the quality of cost-effectiveness analyses, which may be used to design or optimize strategies for primary prevention, early detection and disease management.

In this retrospective study, we, thus, sought to assess the sex- and age-specific treatment costs for CRC over time in the German statutory health care system, firstly in subjects newly diagnosed with CRC and secondly in subjects who died of CRC.

## 2. Methods

### 2.1. Data Source

For this study, we used healthcare insurance claims data obtained from the Vilua Healthcare research database. The database contains entries for approximately 3.5 million individuals across Germany, which represents approximately 4% of the German population covered by statutory health insurance (Gesetzliche Krankenversicherung; GKV). Information on in- and outpatient diagnoses, medication, costs, procedures, diagnosis-related groups (DRGs) and demographics is regularly collected and inspected for outliers, errors and changes over time. As all data were anonymized and deidentified prior to analysis, no approval by an institutional review board or informed consent was required. We followed the Strengthening the Reporting of Observational Studies in Epidemiology (STROBE) guidelines [15].

### 2.2. Study Population

This analysis was performed on consecutive insurance years from 2007 to 2016 in subjects > 18 years of age with the presence of an ICD-10-GM (10th revision of the International Statistical Classification of Diseases and Related Health Problems) code for CRC (C18.*–C20.*) [16]. We included subjects who were at least five years continuously insured within the study period and excluded subjects with another primary cancer diagnoses in any year of the whole observation period. Furthermore, patients were only included if they either had a colonoscopy or a surgery within a year prior or after the date of diagnosis.

To reflect the initial and end-of-life phases of care, we assessed two groups of subjects:Subjects newly diagnosed with CRC;Subjects who died of CRC.

In population (1), to ensure that cost estimates reflected state-of-the-art treatments, we included subjects with new CRC diagnoses (two consecutive outpatient diagnoses or at least one inpatient diagnosis) from the years 2012 onwards who did not die during the observation period. Subjects with an ICD-10 code for CRC within the five years preceding the diagnosis were excluded to ensure that covered cancer cases were truly de novo. Population (2) included all subjects with prevalent CRC from 2007 onwards who died in 2012 or later to reflect recent developments in treatment schemes. Notably, population (2) also included individuals with new CRC diagnoses and death between 2012 and 2016, who, in the absence of death by CRC, would have been allocated to population (1). To assess the influence of such individuals with poor prognosis, we excluded these subjects in a sensitivity analysis.

### 2.3. Analysis of Cost Positions

We defined a positive list of all potentially CRC-related cost positions, broken down into inpatient, outpatient and medication costs, and assessed these costs for all populations. To be considered CRC-related, all inpatient, outpatient and medication costs required coding of CRC as primary or secondary diagnosis. Additionally, total healthcare costs, e.g., all incurring costs regardless of the origin, were assessed. In terms of costing, the Vilua Research Database contains per-case cost position entries for inpatient, outpatient or medication intake. Cost positions was then multiplied with the actual cost as incurred and reported by the collaborating health insurance in their corresponding year of origin (i.e., for this study, the years 2012–2016).

Inpatient costs, including all partial and full hospital procedures, such as surgeries, diagnostic procedures, provision of medicines, remedies and aids during the hospital stay, nursing care and expenses for room and board, were based on the German DRG model, which simplified works by case-based flat rates depending on the medical condition. Outpatient treatment costs were based on the Uniform Assessment Standard (German: Einheitlicher Bewertungsmaßstab, EBM), a catalogue which regulates invoicing for all ambulatory procedures. For this study, ambulatory medication costs were assessed separately (based on the ATC nomenclature), and stratified by chemotherapeutic drugs (e.g., 5-capecitabin), innovative medicines (e.g., biologics, such as bevacizumab), possibly CRC-related medication and further medicines (explicitly not related to CRC). Hospital and outpatient costs were considered CRC-related when the ICD-10 diagnosis for CRC was coded in the hospital stay (either primary or secondary diagnosis) or the outpatient case (only confirmed (“gesicherte”)) diagnoses. Further details are provided in Appendix A.

Of note, the interpretation of health care costs typically depends on whose perspective is considered (e.g., patient, payer or provider), with varying terms and definitions [17]. For this study, we used the costs as reimbursed to providers, i.e., without additional copayments by insured persons or additional discounts or rebate contracts of the insurance companies.

In newly diagnosed patients (population 1), annual costs were assessed for up to five years after first diagnosis (i.e., from date of diagnosis to end of year 1–5 after diagnosis (not calendar years)), and in patients who died of CRC (population 2), for up to five years prior to death (i.e., 1–5 years prior to actual date of death). For each population, all cost positions were assessed separately for men and women and stratified by age (<70 years vs. ≥70 years). We then calculated the mean and median costs with associated parameters of dispersion, both in terms of total CRC-related costs as well as separately for each cost category. For the total CRC-related costs in the first year after diagnosis and the final year before death, where the highest costs were to be expected [6,7], differences by sex and age were tested with unpaired two-sample Wilcoxon rank sum tests.

## 3. Results

### 3.1. Initial Phase of Care

For patients with newly detected cancers (population 1), we identified 1748, 1275, 503 and 359 subjects (females, 51–52%) with follow-up data for 1, 2, 3 and 4 years, respectively (see Appendix A for patient flowcharts). Only three individuals could be identified with follow-up data for the fifth year after diagnosis (not further reported) (Table 1).

Boxplots revealed a strong dispersion of data. In individual observations, CRC treatment costs exceeded EUR 150,000, EUR 110,000, EUR 60,000 and EUR 50,000 in the first, second, third and fourth years after diagnosis, respectively, while corresponding median and average costs were notably lower (Appendix A). Average CRC treatment costs were highest in the year of diagnosis (EUR 16,375–16,450 in men, EUR 10,071–13,250 in women) and declined substantially in the subsequent years (Figure 1). Costs tended to be higher for men vs. women, most markedly in the first year of diagnosis for those <70 years (Wilcoxon rank sum test statistic Z = 5.74, *p*-value < 0.001). A breakdown by cost category revealed higher inpatient costs in men to be the key driver for this pattern (Figure 2). With respect to age, no consistent pattern of differences was observed. Average total healthcare costs were mostly marginally higher than CRC-related costs (see more details in Appendix A).

### 3.2. Terminal Phase of Care

For population 2, we identified 1117, 821, 501, 462 and 279 subjects with data for years 1–5 prior to death, respectively. The proportion of women was substantially lower in those <70 years of age (26–35%) and slightly higher in those ≥70 years (53–54%).

The distribution of CRC treatment costs in the terminal phase of care was also strongly skewed, with maximum costs for individual patients exceeding EUR 250,000, EUR 100,000, EUR 100,000, EUR 150,000 and EUR 60,000 1–5 years prior to death, respectively. Average treatment costs followed a growth trajectory towards the end-of-life, with differences by sex and age. In the final year of life, costs peaked at EUR 34,351 in men and EUR 31,417 in women < 70 years, and at EUR 14,463 in men and EUR 9930 in women ≥ 70 years. In contrast to the years after a new diagnosis (which peaked in the first year and substantially declined in subsequent years), costs in the terminal phase of care were consistently on a comparably high level (relative to the final year of life), even several years before death (Table 2).

Differences by sex were most marked for those ≥70 years, with average treatment costs in men vs. women being 41% and 46% higher in the penultimate and last year, respectively. The key determinant for this difference was costs for innovative medicines (Figure 3).

Differences by age, observed both for men and women, were even more pronounced than the differences by sex. Average total CRC treatments for those <70 years were at least twice the costs for those ≥70 years in each year of follow-up. These strong differences were mostly driven by substantially higher costs in the younger age group for inpatient care (approximately two times higher costs), chemotherapeutic drugs as well as novel medicines (approximately three times higher costs) (Figure 3). Notably, though lower as compared to the younger age group, treatment costs for older patients in the terminal phase were still substantially higher than treatment costs in the initial phase of care.

Average total healthcare costs were consistently several thousand EUR higher than CRC-related costs, with the largest differences observed in the final year of death. This was mostly driven by inpatient costs not unambiguously identifiable as related to a CRC treatment (Appendix A).

A sensitivity analysis excluding individuals with poor prognosis (diagnosed and died within 2012–2016) found lower estimates, but was overall consistent with the patterns seen in the main analyses (Appendix A). Separate analyses further stratifying by location (colon and rectosigmoid junction vs. rectum) did not suggest marked differences in mean treatment costs, though costs in the initial phase of care were slightly higher for rectum cancers (data not shown).

## 4. Discussion

Based on claims data, this study provided detailed sex- and age-specific estimates on average CRC treatment costs over time in Germany, both in newly detected cases as well as prior to death. In those newly diagnosed, costs were highest in the first year after diagnosis and dropped steeply in the following years. Costs in the terminal phase of care were consistently on a high level even several years preceding death, mostly driven by high inpatient and medication costs, and substantially higher as compared to the initial phase of care. We observed a consistent pattern of higher costs in men versus women, most markedly in the first year of diagnosis and the penultimate and final year of life. Finally, the costs prior to death varied strongly by age, with average costs to treat individuals < 70 years at age of death amounting to at least twice the costs of those ≥70 years.

## 5. Findings in Context

Our findings were in agreement with previous evidence, in that highest costs arose in the first and final year(s) of care [6,7,11,18,19,20]. Typically, distributions of CRC treatment costs tended either to be ‘U-shaped’ (i.e., similarly high costs for initial and end-of-life phases) or ‘J-shaped’ (i.e., high costs in the initial phase but even higher costs in the end-of-life phase). In analogy, the cost estimates of our study may be regarded as following a ‘J-shaped with tail’ form, implying that costs were considerable even several years prior to death. This distribution may be attributed to continuous and ever more intensive care with the gradual worsening of the disease towards the end of life, also involving particularly cost-intensive therapeutic options such as innovative drugs, as seen in the cost breakdown of our data.

Our cost estimates for the initial and final 12 months of care were notably lower than those recently reported in a claim-based study from Saxony, one of the Eastern states of Germany [6]. This may be attributed to several methodological differences. Firstly, we used more restrictive criteria upon selecting the study population as compared to previous studies, for instance, regarding allowed concomitant primary cancers other than CRC. Secondly, we applied a positive list of costs, i.e., by only including costs of procedures clearly related to CRC. As other cost drivers unrelated to CRC were, therefore, excluded, we believe that our more conservative, and less pragmatic, approach led to estimates more specifically assessing CRC treatment costs.

Thirdly, the initial and end-of-life phases in previous studies quantifying CRC treatment costs in Germany covered only 12-month periods [6,7]. With the rise of modern drug therapies, which were shown to prolong the overall survival [21,22] and may be indicated across several lines of therapy [23,24,25], costs are expected to become more substantial throughout several years of palliative care, and not limited to the final year of life alone. A key strength of our study was that we addressed this evidence gap by quantifying the costs up to five years after diagnosis and prior to death, which likely resulted in a more accurate reflection of treatment costs over time. However, in line with previous studies [6,7,8,11], precedence was given to the end-of-life over the initial phase of care. This implied that costs in the initial phase of care were censored for patients who survived less than five years. As suggested by our sensitivity analysis, including these patients in the initial phase of care instead would likely have resulted in higher estimates more comparable to those previously published.

Notably, screening for CRC is generally accepted as cost-effective or even cost-saving [26,27] by removing precancerous lesions, thereby preventing CRC in the first place [28]. The preventive potential is tremendous, with estimated incidence reductions of 60–70% for the use versus nonuse of screening colonoscopy [28,29]. However, most cost-effectiveness analyses have focused on non-European populations [26], and evidence for Germany is particularly scarce [4,5,6]. Previous findings may not be directly transferable due to differences in healthcare systems, population characteristics, demographic trends as well as country-specific screening offers. More importantly, several (rather unique) sex- and age-specific options are currently offered in Germany, but it remains unclear how their cost-effectiveness compares to each other, and whether there may be more cost-effective alternatives [30]. While out of scope for the here presented study, our findings could be highly useful in informing such dedicated analyses, thereby contributing to optimizing strategies to screen for CRC.

Finally, the observed differences by sex and age warrant further investigation. Similar differences were also seen in previous studies, mostly in US American populations [8,9,10,11,13]. Most of these differences may be attributable to more intensive care for patients diagnosed at higher stages of disease [11]. Unfortunately, the nature of the claims data used in our study did not allow to evaluate the role of stages. However, stage distribution was found to be very similar among male and female CRC patients in Germany, and is, therefore, unlikely to explain the major differences by sex [3]. Generally, treatment guidelines do not recommend age- or sex-specific treatment strategies, but emphasize that patients need to be at good general condition for certain treatments [23,25]. However, it was shown that younger cancer patients tended to seek more aggressive surgical care and more adjuvant treatment than older cancer patients [8,12]. It appears, therefore, plausible that some elderly patients may be too fragile to receive burdensome surgeries or aggressive drug–medicinal interventions, thus, decreasing average treatment costs. Further possible explanatory factors beyond the scope of our study include previous treatments, the site of disease, tumor mutational status and patient preferences [23,25].

## 6. Limitations

The limitations of this study included the small sample size, the retrospective nature, the limited observation period and the reliance on insurance data, implying that incorrect reporting could not be ruled out. For instance, the quality of death information in our dataset was not guaranteed, and CRC deaths could not be reliably separated from deaths for other reasons. However, potential bias by costs other than for CRC treatment was minimized by specifically designed eligibility criteria (e.g., excluding patients with other primary cancer diagnoses) as well as by a positive list of costs excluding non-CRC-related positions.

Furthermore, subjects who developed metastatic disease may have been slightly underrepresented in our sample, as distant metastases are sometimes incorrectly coded as new primary cancer, which were excluded in our study. Our cost estimates may, therefore, have been slightly underestimated, as metastatic cancers require more intensive treatment.

Treatment costs may have been further underestimated by the increased use of immuno-oncological drugs, such as nivolumab or pembrolizumab, which were only authorized in early 2021 for specific patients with metastatic CRC in most European countries, including Germany. As these rather cost-intensive drugs were not yet part of standard clinical practice for CRC during the observation period of this study (2007–2016), possibly associated spikes in treatment costs since their market authorization were not reflected in our estimates.

As well, the relatively small number of patients included in our analyses, in particular for >1 year after diagnosis or >1 year prior to death, implied that the generalizability of our findings may need to be interpreted with caution. Ideally, costs should be assessed with a prospectively planned incidence-based approach, e.g., with a long-term survival follow-up of a cohort of incident CRC cases linked to health insurance data, which may allow to compare costs from diagnosis to end of life. However, as this approach is highly complex [7,31], the here presented retrospective analysis of healthcare claims data may represent a simpler, but also much more feasible, alternative to generate insights on CRC costs over time. Unfortunately, given the restrictions in sample size and relatively short durations of follow-up for most patients in our sample, a pure survival-based analysis was not in the scope of our study. Therefore, more high-quality studies on CRC treatment costs over time are needed to substantiate and possibly refine our findings. Apart from a survival-based approach, this may also extend to further separating patients by mode of detection at diagnosis, e.g., screening versus symptoms. We abstained from such further stratification as only a very small number of screen-detected cancers could be identified in our sample, essentially impeding any meaningful interpretation.

Finally, while we stratified for sex and age, a large heterogeneity between these predefined subgroups, e.g., with respect to the distribution of comorbidities, could not be ruled out, which caused intergroup comparisons to be more difficult. We purposely abstained from matching individuals between these groups as this would have further reduced the already small sample size. On the one hand, the use of a positive list of costs likely limited the impact of the heterogeneity between groups, as strictly only CRC-related costs were considered. On the other hand, this approach may also imply that some expenses which in fact occurred for the treatment of CRC were missed, e.g., if a procedure was incorrectly claimed. Though we believe this to have occurred very rarely, it may contribute to a possible underestimation of costs.

## 7. Conclusions

Once clinically manifest, CRC causes substantial treatment costs over time, particularly in the palliative care setting. Observed higher costs in men versus women as well as strong differences by age in the terminal phase of care warrant further investigation, which should also additionally address indirect costs.

## Figures and Tables

**Figure 1 cancers-14-03836-f001:**
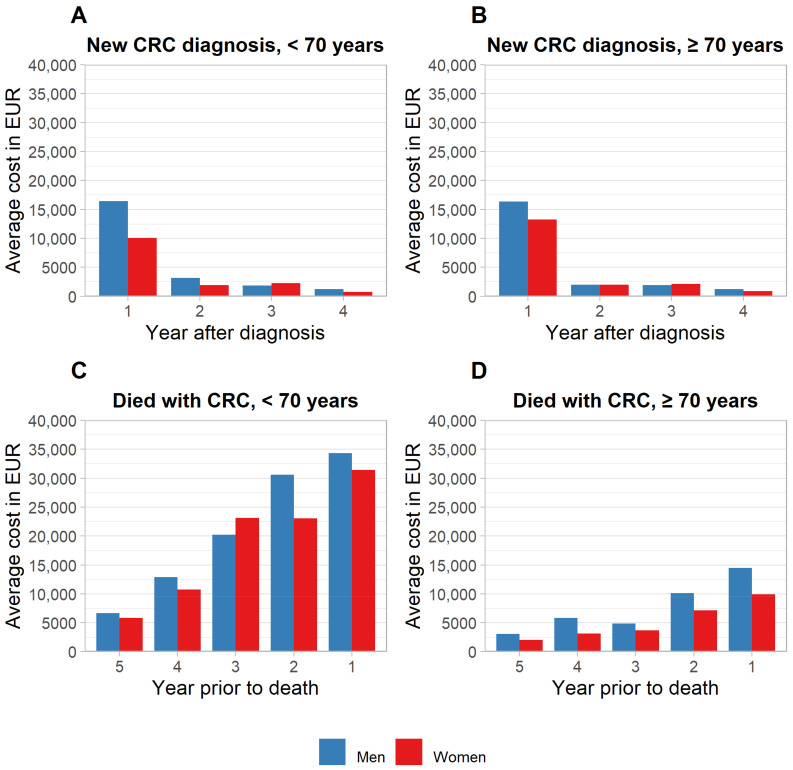
Average total treatment costs of colorectal cancer after new diagnosis and prior to death, stratified by sex and age. (**A**,**B**) Newly diagnosed cases. (**C**,**D**) Subjects who died of CRC. CRC, colorectal cancer.

**Figure 2 cancers-14-03836-f002:**
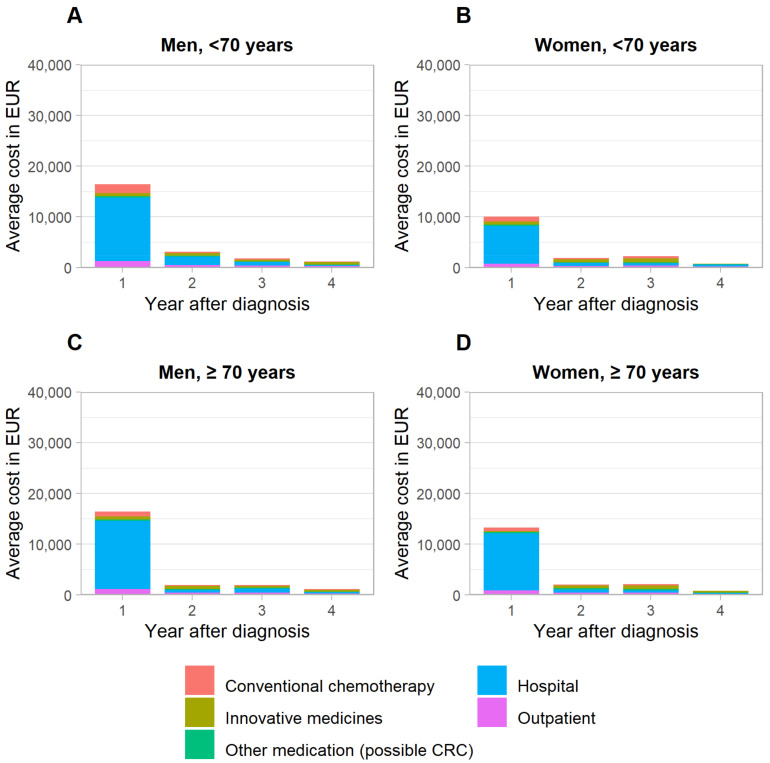
Breakdown of average treatment costs in newly detected colorectal cancer cases, stratified by sex and age. CRC, colorectal cancer. (**A**): Men, <70 years, (**B**): Women, <70 years, (**C**): Men, ≥70 years, (**D**): Women, ≥70 years.

**Figure 3 cancers-14-03836-f003:**
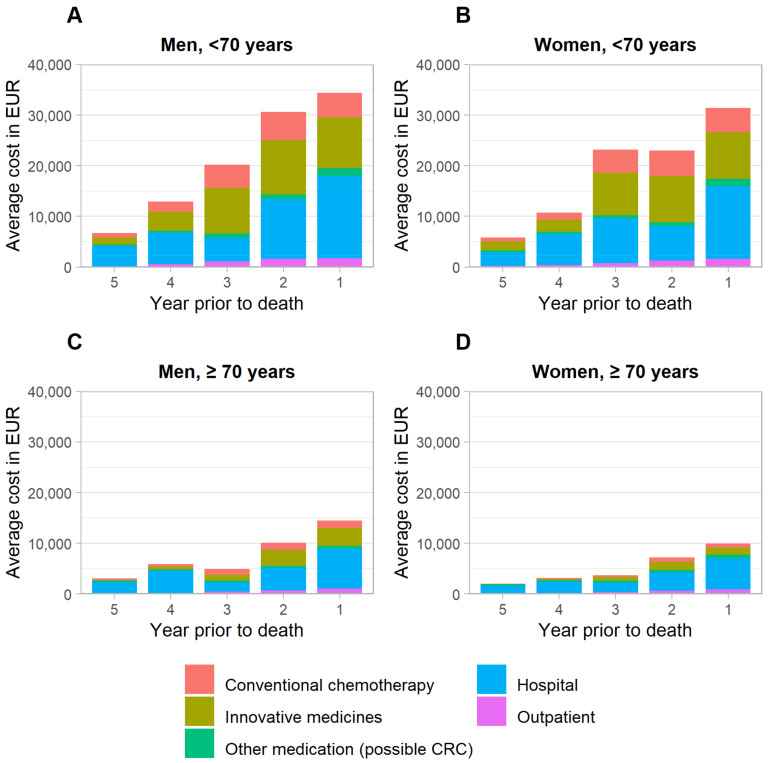
Breakdown of average treatment costs prior to death, stratified by sex and age. CRC, colorectal cancer. (**A**): Men, <70 years, (**B**): Women, <70 years, (**C**): Men, ≥70 years, (**D**): Women, ≥70 years.

**Table 1 cancers-14-03836-t001:** Inpatient, outpatient and medication costs, as well as total healthcare costs 1–5 years after diagnosis, stratified by sex and age.

Year after Diagnosis	No of Patients	Median Age (Years)	Average CRC-Related Costs in EUR				Average Total Healthcare Costs in EUR
			Hospital	Ambulatory	Medication	Total Costs	
			**Men**				
**<70 age**							
1	486	59	12,566	1235	2648	16,450	20,245
2	336	60	1677	449	1022	3147	6435
3	119	61	634	437	738	1809	3879
4	79	62	208	287	708	1203	2374
**≥70 age**							
1	360	75	13,399	1187	1789	16,375 ^1^	21,210
2	285	76	493	491	1014	1998	6240
3	120	77	815	470	635	1920	5832
4	94	78	273	323	598	1195	3450
			**Women**				
**<70 age**							
1	411	58	7519	716	1836	10,071 ^2^	13,109
2	283	59	589	324	986	1900	3938
3	91	59	451	375	1399	2226	4831
4	55	62	331	201	233	766	2211
**≥70 age**							
1	491	78	11,177	913	1160	13,250 ^3,4^	17,728
2	371	78	635	449	925	2009	6392
3	173	78	447	481	1195	2122	5477
4	131	79	110	244	487	841	2574

Costs as actually incurring in the period 2012–2016. ^1,2,3,4^ Explorative Wilcoxon rank sum tests for differences in total cost in the first year after diagnosis. ^1^ By age group, men. Test statistic, <70 years vs. ≥70 years: z = 1.95; *p*-value = 0.051. ^2^ By sex, <70 years. Test statistic, men vs. women: z = 5.74; *p*-value = <0.001. ^3^ By age group, women. Test statistic, <70 years vs. ≥70 years: z = 7.04; *p*-value = <0.001. ^4^ By sex, ≥70 years. Test statistic, men vs. women: z = 2.43; *p*-value = 0.015.

**Table 2 cancers-14-03836-t002:** Inpatient, outpatient and medication costs 1–5 years prior to death, stratified by sex and age.

Year before Death	No of Patients	Median Age (Years)	Average CRC-Related Costs in EUR				Average Total Healthcare Costs in EUR
			Hospital	Ambulatory	Medication	Total	
			**Men**				
**<70 age**							
5	35	62	4007	152	2538	6696	12,265
4	64	61	6197	557	6137	12,892	16,566
3	70	61	4633	1124	14,436	20,193	26,804
2	117	60	11,852	1607	17,150	30,609	38,094
1	165	60	16,174	1736	16,441	34,351	45,460
**≥70 age**							
5	108	81	2182	109	779	3070	6652
4	176	80	4328	236	1310	5874	10,174
3	191	81	1831	443	2632	4907	10,563
2	303	81	4447	743	4919	10,109	15,462
1	393	81	7870	1073	5520	14,463 ^1^	25,387
			**Women**				
**<70 age**							
5	15	64	2668	187	2980	5835	9073
4	23	62	6007	436	4317	10,759	15,040
3	25	63	8845	742	13,564	23,151	26,426
2	56	62	6821	1296	14,924	23,041	30,168
1	90	60	14,397	1603	15,417	31,417 ^2^	45,312
**≥70 age**							
5	121	82	1514	109	415	2038	4864
4	199	83	2179	212	757	3148	6496
3	215	84	1874	353	1465	3692	7579
2	345	84	3589	663	2927	7180	11,819
1	469	84	6138	917	2875	9930 ^3,4^	19,259

Costs as actually incurring in the period 2007–2016. ^1,2,3,4^ Explorative Wilcoxon rank sum tests for differences in total cost in the last year before death. ^1^ By age group, men. Test statistic, <70 years vs. ≥70 years: z = 9.10; *p*-value = <0.001. ^2^ By sex, < 70 years. Test statistic, men vs. women: z = 0.55; *p*-value = <0.585. ^3^ By age group, women. Test statistic, <70 years vs. ≥70 years: z = 9.06; *p*-value = <0.001. ^4^ By sex, ≥70 years. Test statistic, men vs. women: z = 1.26; *p*-value = 0.209.

## Data Availability

Code and material: all analyses relevant to the study were included in the article or uploaded as Appendix A.

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
