# Peer review of "Treatment Costs of Colorectal Cancer by Sex and Age: Population-Based Study on Health Insurance Data from Germany"

_cancers, 2022, doi:10.3390/cancers14153836_

Round 1

Reviewer 1 Report

The authors analyze the costs of treatments incurred by patients diagnosed with colorectal cancer (CRC) by screening through a database of insurance claims for the treatment. The study is pretty straightforward. They categorized the patients into 2; newly diagnosed and those who died of CRC.

One of the limitations the authors quote is the small sample size. How does the small sample size represent the total costs of treatment in CRC patients in Germany? This would be interesting to know.

It will be good to see a couple of sentences regarding this in the Limitations or the Conclusion sections.

Reviewer 2 Report

The article by Heisser et al. is a timely evaluation of the financial burden of colorectal cancer in different stages in Germany. The higher cost of treatment of colorectal cancer in the first year is easily explained due to surgery costs. In the following years the data is scarce. 

There are some issues that need to be explained before moving on:

1.) How many patients were with rectal cancer? This needs to be explained for the readers due to the higher cost of radiotherapy needed for this category of patients.

2.) Is immunotherapy (Pembrolizumab, Nivolumab +/- Ipilimumab) available for patients with MSI-H/d-MMR in Germany? This will increase considerably the treatment costs if available.

Reviewer 3 Report

I have read carefully the manuscript prepared by your team and I can say that you have submitted a sustained work and a quality analysis of the costs necessary for the treatment of patients with colorectal cancer according to sex and age.
The methods of patient selection and the number of patients included in the study are relevant and sufficient for a quality statistical analysis and the results obtained have statistical significance.
The discussions are pertinent also in the context of the study as well as in comparison with the results of other studies, being explained in detail the differences of results.
The conclusions result from the analysis of the data entered in the study and are consistent with them.
As a result, I recommend publishing the manuscript in its current form.

Reviewer 4 Report

This is a descriptive study of the treatment costs of colorectal cancer based on the health insurance data from Germany. The major limitation of the article is the lack of detailed and in-depth study of mechanism.
